# Roles of Gangliosides in Hypothalamic Control of Energy Balance: New Insights

**DOI:** 10.3390/ijms21155349

**Published:** 2020-07-28

**Authors:** Kei-ichiro Inamori, Jin-ichi Inokuchi

**Affiliations:** Division of Glycopathology, Institute of Molecular Biomembrane and Glycobiology, Tohoku Medical and Pharmaceutical University, Sendai, Miyagi 981-8558, Japan

**Keywords:** ganglioside, glycosphingolipid, leptin receptor signaling, hypothalamic neurons, energy homeostasis

## Abstract

Gangliosides are essential components of cell membranes and are involved in a variety of physiological processes, including cell growth, differentiation, and receptor-mediated signal transduction. They regulate functions of proteins in membrane microdomains, notably receptor tyrosine kinases such as insulin receptor (InsR) and epidermal growth factor receptor (EGFR), through lateral association. Studies during the past two decades using knockout (KO) or pharmacologically inhibited cells, or KO mouse models for glucosylceramide synthase (GCS; *Ugcg*), GM3 synthase (GM3S; *St3gal5*), and GD3 synthase (GD3S; *St8sia1*) have revealed essential roles of gangliosides in hypothalamic control of energy balance. The a-series gangliosides GM1 and GD1a interact with leptin receptor (LepR) and promote LepR signaling through activation of the JAK2/STAT3 pathway. Studies of GM3S KO cells have shown that the extracellular signal-regulated kinase (ERK) pathway, downstream of the LepR signaling pathway, is also modulated by gangliosides. Recent studies have revealed crosstalk between the LepR signaling pathway and other receptor signaling pathways (e.g., InsR and EGFR pathways). Gangliosides thus have the ability to modulate the effects of leptin by regulating functions of such receptors, and by direct interaction with LepR to control signaling.

## 1. Introduction

Gangliosides (glycosphingolipids (GSLs) that contain one or more sialic acids) are essential components of membrane microdomains, and play key roles in a variety of important biological processes, including cell growth, differentiation, and signal transduction [1]. Ganglioside synthesis is initiated by addition of a glucose residue to the common precursor ceramide to form glucosylceramide (GlcCer) in the Golgi. Subsequently, a galactose residue is added to GlcCer to form lactosylceramide (LacCer), the precursor for synthesis of various ganglioside species and other types of GSLs (Figure 1). GM3 synthase (GM3S, encoded by *St3gal5*) is a sialyltransferase that transfers sialic acid residue to LacCer via α2,3-linkage to form GM3 ganglioside. Based on GM3, GD3 synthase (GD3S, encoded by *St8sia1*) transfers sialic acid via α2,8-linkage to form the disialoganglioside GD3, and GM2 synthase (GM2S, encoded by *B4galnt1*) transfers N-acetylgalactosamine (GalNAc) to form GM2 (Figure 1). GM2S can also act on LacCer and GD3 to generate GA2 and GD2, respectively. Gangliosides are involved in functioning of numerous growth factor receptors and hormone receptors, including epidermal growth factor receptor (EGFR), platelet-derived growth factor receptor (PDGFR), vascular endothelial growth factor receptor (VEGFR), hepatocyte growth factor receptor (c-Met), nerve growth factor receptor (TrkA), insulin receptor (InsR), and insulin-like growth factor 1 receptor (IGF1R) [2]. Recent studies have demonstrated the essential roles of gangliosides in hypothalamic control of feeding and energy homeostasis through regulation of leptin receptor (LepR) signaling.

## 2. Leptin Receptor Signaling

Leptin, a 16-kDa peptide hormone produced mainly from adipose tissue, is essential for maintenance of energy homeostasis and body weight [3]. Adiposity is strongly correlated with circulating leptin level. Leptin acts as a transmitter of metabolic information to the hypothalamus, which plays critical roles in regulation of feeding, body weight, and energy expenditure [4]. Several hypothalamic nuclei, including the arcuate nucleus (ARC), paraventricular nucleus (PVN), ventromedial hypothalamus (VMH), and lateral hypothalamic area (LH), are involved in control of energy homeostasis [5]. ARC contains two interconnected groups of neurons that express long-form leptin receptor (LepRb) and is the main site of leptin activity [6]. The proopiomelanocortin (POMC) neurons are satiety-promoting and tonically release α-melanocyte-stimulating hormone (α-MSH; processed from POMC), which binds to melanocortin receptor 4 (MC4R) in PVN and certain other hypothalamic nuclei, and thereby promotes an anorectic effect and energy expenditure. The agouti-related peptide (AgRP) neurons are hunger-promoting and release (i) AgRP, which competes with α-MSH for MC4R binding in a coordinated fashion to regulate feeding and energy balance, and (ii) neuropeptide Y (NPY) and γ-aminobutyric acid (GABA) for regulation of energy balance. LepRb is expressed in both the above groups of neurons. However, postprandially increased circulating leptin level stimulates POMC neurons (with consequent α-MSH release and inhibition of feeding) but inhibits AgRP neurons. Elevated leptin level typically generates a strong signal that functions to prevent obesity, however, such effect is weak or disrupted in already-obese subjects. Obese rodents and humans display hyperleptinemia and do not respond substantially to exogenous leptin administration [7,8].

Signaling from leptin is generated via binding to its receptor LepR, a transmembrane protein belonging to the class I cytokine receptor family [6]. Mice that lack functional leptin or LepR (*ob/ob*, *db/db*) are hyperphagic and display severe obesity with hyperglycemia and insulin resistance [9]. Six LepR isoforms (termed LepRa through LepRf) present in mice are generated from a single *LepR* gene by alternative splicing or ectodomain shedding [10,11]. The six LepR isoforms are assigned to three subtypes: one soluble form (LepRe), four short forms (LepRa, LepRc, LepRd, LepRf), and one long form (LepRb). The six isoforms have in common an N-terminal extracellular domain capable of binding leptin but have differing C-terminal cytoplasmic domains. The cytoplasmic domain of LepRb (not the other isoforms) contains three conserved tyrosine (Tyr) residues necessary for efficient leptin-mediated signaling. LepRb is highly expressed in brain areas involved in the control of feeding and energy expenditure [12]. LepRb itself has no intrinsic kinase activity, but it generates an intracellular signal through binding to Janus kinase 2 (JAK2) [13]. Leptin binding to LepRb results in conformational change of the receptor, followed by JAK2 phosphorylation. Activated JAK2 phosphorylates three Tyr residues in LepRb (Tyr^985^, Tyr^1077^, Tyr^1138^), and each of the phospho-Tyr (p-Tyr) residues recruits specific Src homology 2 (SH2) domain-containing proteins (Figure 2) [14].

The JAK-STAT (signal transducers and activators of transcription) pathway is the best-studied signal transduction pathway activated by leptin [14,15]. p-Tyr^1138^ is the binding site for the SH2 domain of STAT3. STAT3 is subsequently phosphorylated by JAK2 and translocated to the nucleus as a dimer to initiate expression of target genes, such as suppressor of cytokine signaling 3 (SOCS3), which acts as a negative-feedback regulator of the JAK-STAT pathway. SOCS3 is a member of a large family of cytokine-inducible inhibitors of signaling, and its gene expression is induced by leptin [16]. SOCS3 binds to p-Tyr^985^ of LepRb and mediates negative feedback inhibition of LepRb signaling by inhibiting JAK2 activation [17,18]. This pathway is essential for the anti-obesity activity of leptin in the brain. A substitution mutation of Tyr^1138^ to serine abolishes STAT3 binding, and mice with this mutation (*s/s* mice) display hyperphagia and obesity similar to those of *db/db* mice. In comparison with *db/db* mice, on the other hand, *s/s* mice display improved insulin sensitivity, glycemic control, linear growth, and normal fertility [19,20,21]. Mice with neuronal-specific deletion of STAT3 also display severe obesity, decreased linear growth, and infertility, similarly to *db/db* mice. LepRb-expressing neuron-specific deletion of STAT3 also results in the obese phenotype; however, the mice are fertile and display enhanced linear growth [22,23]. These observations indicate that STAT3 is a key mediator of leptin activity and essential for energy balance but is not required for growth or reproduction.

Besides the LepRb-STAT3 pathway, leptin activates STAT5 through phosphorylation of Tyr^1077^. Mice with a substitution mutation of Tyr^1077^ to phenylalanine show only minor increases in body weight and adiposity [24]. Mice with neuronal deletion of STAT5 using nestin-cre develop severe obesity, but specific STAT5 deletion in LepRb-expressing neurons does not result in the obese phenotype [25,26], suggesting that the LepR-STAT5 pathway does not play an essential role in body weight regulation, and that STAT5 may mediate other cytokine signals for regulation of energy balance.

The LepRb-SHP2 (SH2-containing protein Tyr phosphatase 2)-ERK (extracellular signal-regulated kinase) pathway is also involved in the effect of leptin on energy balance through phosphorylation of Tyr^985^, which serves as a binding site for the SH2 domain of SHP2, and also for SOCS3. SHP2 is phosphorylated by JAK2 and recruits Grb2 (growth factor receptor-bound protein 2), the adaptor protein that mediates ERK activation. SHP2 is thus a positive regulator of leptin-mediated ERK activation, and its phosphatase activity is necessary for the pathway [27]. Pharmacological blockade of ERK in the hypothalamus reverses the anorectic and thermogenic sympathetic effects of leptin [28]. Mice with POMC neuron-specific or neuronal-specific deletion of SHP2 display the obese phenotype [29,30]. These findings indicate the importance of the LepRb-SHP2-ERK pathway for regulation of energy balance. In contrast, mice with a substitution mutation of Tyr^985^ are lean and (particularly in females) resistant to diet-induced obesity (DIO) [31]. These Tyr^985^ mutant mice display reduced hypothalamic expression of *Agrp* and *Npy*, and increased leptin sensitivity resulting from suppressed SOCS3 binding to LepRb, reflecting the primary role of SOCS3 in feedback inhibition of LepRb. In regard to the LepRb-SHP2-ERK pathway, the role of SOCS3 is not reflected by the phenotype of Tyr^985^ mutant mice. SHP2-ERK is involved in several different signaling pathways, and its specific contribution to LepRb signaling is therefore difficult to clarify.

Other signaling pathways are also involved in leptin activity. Leptin activates phosphatidylinositol 3-kinase (PI3K) through phosphorylation of insulin receptor kinase substrate 2 (IRS2). Systemic administration of leptin in rats activates PI3K in the hypothalamus, and intracerebroventricular infusion of PI3K inhibitors blocks leptin-induced anorexia [32]. Specific deletion of IRS2 in LepRb-expressing neurons results in obesity and insulin resistance [33]. IRS2 in LepRb neurons is crucial for metabolic signaling, however, requirement of this pathway is independent of leptin activity in the neurons. To date, no specific p-Tyr on LepRb has been identified as the binding motif for their recruitment. The PI3K pathway is shared with other receptors, including InsR. Specific roles of PI3K in leptin signaling are thus fairly complex, and evaluation of the specific contribution of this pathway to leptin’s effect on energy homeostasis is difficult.

SH2B1, a binding protein and strong positive regulator of JAK2, binds to IRS2 and links JAK2 to activation of the PI3K pathway by leptin in cultured cells [34]. SH2B1 null mice are hyperphagic and severely obese, and this phenotype is blocked by neuron-specific restoration of SH2B1 [35,36,37]. Rui’s group recently demonstrated that LepRb-expressing neuron-specific SH2B1 null mice develop insulin resistance and liver steatosis [38]. Despite fairly normal food intake and energy expenditure in these mice, leptin-induced sympathetic nerve activation was suppressed, resulting in dysfunctional brown adipose tissue and reduced core body temperature. SH2B1 thus appears to act as an endogenous sensitizer of leptin activity, most likely through promotion of JAK2 activation. However, SH2B1 also mediates other types of signals, notably insulin signaling, and it is therefore difficult to clarify its specific role in leptin signaling. It may enhance the abilities of both leptin and insulin to activate sympathetic nerves and/or increase energy expenditure.

A Tyr-phosphorylation-independent LepRb signaling pathway was described by Liu’s group [39]. Mice with substitutions of all three Tyr residues in LepRb (Tyr^985^, Tyr^1077^, Tyr^1138^) by phenylalanine (Y123F mice) developed obesity but displayed reduced adiposity and hyperphagia, enhanced glucose homeostasis, and sustained fertility in contrast to *db/db* mice, indicating a Tyr-independent mechanism of LepRb signaling in control of energy balance [39]. A truncation mutant of LepRb, which retained the ability to activate JAK2 but lacked Tyr^985^, Tyr^1077^, and Tyr^1138^, resulted in a phenotype similar to that of *db/db* mice in regard to energy balance, obesity, and infertility [40]. LepRb-phosphorylation-independent JAK2 signaling thus appears insufficient to mediate the improved phenotype observed in Y123F mice relative to *db/db* mice. The above findings indicate the existence of some other yet-to-be-identified signal, independent of Tyr phosphorylation. LepRb regions that mediate such signals were recently identified by generating a series of mutant mice carrying LepRb truncation mutations [41]. LepRb sequences between residues 921 and 960, which contain no Tyr residues, mediated a signal involved in the control of feeding and energy balance. A different sequence comprising residues 1013 through 1053, which also did not contain Tyr, mediated an inhibitory signal. Whether specific molecules are recruited to these regions, and whether certain regions are required for other cellular functions, remains unknown. It is possible that the role of a given region is redundant with that of some other LepRb sequence. Precise sequences that mediate specific signals will be identified in future studies.

## 3. Suppression of Leptin Signaling

Hyperphagia and obesity can be normalized by administration of exogenous leptin to leptin-deficient human subjects or *ob/ob* mice. On the other hand, leptin administration to obese subjects or animals, in whom circulating leptin levels are typically elevated, does not substantially reduce food intake or body weight. This phenomenon has been attributed to “leptin resistance”, a state in which elevated or exogenously administered leptin is insufficient to reduce feeding and body weight. Administration of leptin receptor antagonist results in comparable increases of feeding and body weight in lean and hyperleptinemic DIO mice, indicating that endogenous leptin suppresses feeding even in the obese mice [42]. These findings suggest that in the obesity that typically accompanies hyperleptinemia, leptin activity reaches a defined maximal value. Continued elevation of leptin beyond this value has essentially no additional effect and does not suppress feeding.

Chronically elevated levels of circulating leptin in obese rodents and humans activate several pathways that lead to a negative feedback mechanism, resulting in suppression of LepRb signaling. Leptin signaling through LepRb Tyr^1138^ and the JAK2-STAT3 pathway promotes SOCS3 expression (Section 2). SOCS3 binds to p-Tyr^985^ of LepRb to mediate a negative feedback loop of LepRb signaling by inhibiting activation of JAK2. SOCS3 expression in ARC is elevated in DIO mice and in a hyperleptinemic obese mouse model *Ay/a*. Intraperitoneal leptin administration rapidly induces SOCS3 expression in hypothalamus of *ob/ob* mice, but not of *db/db* mice [43]. Mice with neuronal deletion of *Socs3* using either nestin-cre or synapsin-cre display increased leptin sensitivity and resistance to DIO, indicating that SOCS3 is a physiological negative regulator of LepRb signaling [44]. On the other hand, deletion or overexpression of SOCS3 in LepRb-expressing cells of mouse hypothalamus did not result in DIO, suggesting that the mechanism whereby SOCS3 inhibits LepRb signaling in vivo is not as straightforward as initially considered [45,46]. Expression of other proteins that modulate leptin sensitivity (e.g., STAT3, protein Tyr phosphatases) in these models may undergo compensatory changes, and SOCS3 may have neuron-type-specific effects on energy balance. LepRb-expressing cell-specific *Socs3* null mice placed on a high-fat diet (HFD) did not display notable increases in body weight. However, they showed increased leptin sensitivity in the hypothalamus and did not develop diet-induced insulin resistance, suggesting that regulation of LepRb signaling by SOCS3 mediates diet-induced changes of glycemic control [45]. These null mice displayed reduced food intake and weight regain after fasting, based on low transcription of orexigenic neuropeptides, indicating that SOCS3 regulates fasting-induced hyperphagia and weight regain [47].

Leptin signaling is also suppressed by other proteins, particularly protein Tyr phosphatases (PTPs). Leptin signaling depends on JAK2 phosphorylation and subsequent phosphorylation of Tyr residues on LepRb and STAT3; therefore, PTPs that act on JAK2, STAT3, or LepRb may regulate LepRb signaling. PTP1B, a non-receptor Tyr phosphatase expressed in hypothalamus that inhibits insulin signaling by dephosphorylating IRS, suppresses LepRb signaling by directly dephosphorylating JAK2 [48]. PTP1B-null mice are lean, hypersensitive to leptin, and resistant to DIO [49,50]. Mice with neuronal deletion or POMC-neuron-specific deletion of PTP1B display increased leptin sensitivity and energy expenditure, and reduced DIO susceptibility, suggesting that increased PTP1B expression in obesity suppresses leptin signaling [29,51]. RPTPε, a transmembrane receptor-type Tyr phosphatase, modulates ERK signaling by inhibiting ERK1/ERK2 kinase activity [52], and suppresses LepRb signaling by dephosphorylating JAK2 [53]. RPTPε activity is elevated in obese mice. Leptin stimulation induces phosphorylation of hypothalamic RPTPε at C-terminal Tyr^695^, thereby downregulating LepRb signaling through a negative feedback mechanism. RPTPε-null mice are leptin-sensitive and protected from HFD-induced obesity. 

T cell PTP (TCPTP), another PTP closely related to PTP1B, is elevated in the hypothalamus of DIO mice and suppresses leptin activity [54]. Its substrate specificity differs from that of PTP1B. TCPTP dephosphorylates JAK1 and JAK3, whereas PTP1B dephosphorylates JAK2 [55]. An alternatively spliced isoform of TCPTP (TC45) is localized to the nucleus, and TCPTP has been suggested to regulate leptin signaling via dephosphorylation of STAT3. Mice with neuronal deletion of TCPTP display enhanced leptin sensitivity and DIO resistance. Mice with TCPTP/PTP1B double knockout (KO) in neuronal cells show additive effects in DIO resistance, suggesting that TCPTP and PTP1B may act in a coordinated fashion within a given neuronal population and inhibit distinct leptin signaling molecules (JAK2 and STAT3, respectively) [56].

## 4. Ganglioside-Regulated Receptor Signaling

Gangliosides are synthesized by sequential addition of sugar units in Golgi and transported to the outer leaflet of the plasma membrane, where they play essential roles in the regulation of a subset of growth factor and hormone receptors. In membrane microdomains and lipid rafts, gangliosides associate laterally with a variety of components, including other sphingolipids, cholesterol, glycosylphosphatidylinositol (GPI)-anchored proteins, and a subset of transmembrane receptors [57]. Studies during the past decade indicate that most microdomains/rafts are very small (nanoscale size) and highly dynamic. Kiso’s group observed dynamic exchange of gangliosides between raft domains and bulk domains [58].

Growth factor receptors play essential roles in various cellular functions, and many of them use receptor Tyr kinases for intracellular signaling. Early studies in the 1980s showed that exogenous addition of gangliosides to culture medium inhibited cell proliferation induced by EGF or PDGF [59], and that GM3 inhibited EGF-induced autophosphorylation of EGFR [60]. GM3 allosterically inhibits EGFR Tyr kinase activity by interacting with a specific membrane proximal region and also with N-linked GlcNAc termini of the receptor [61,62]. Other receptor Tyr kinases including VEGFR, c-Met, TrkA, and IGF1R (see Section 1) have been shown to be regulated (activated in some cases, inhibited in others) by various gangliosides [63,64,65]. In general, monosialogangliosides (particularly GM3) inhibit receptor Tyr kinases, whereas b-series gangliosides (i.e., tandem disialogangliosides) activate them. The case of GD1a, a disialoganglioside belonging to a-series, is more complex. In a study using a highly metastatic mouse osteosarcoma cell line FBJ-LL, GD1a inhibited hepatocyte growth factor-induced phosphorylation of c-Met, and cell motility [66]. In a study using human neuroblastoma cell line NBL-W, GD1a (and also GM3, GM1, and GT1b) inhibited EGFR phosphorylation and cell proliferation [67]. In normal human dermal fibroblasts, on the other hand, GD1a promoted EGFR dimerization in the absence of EGF, thereby enhancing ligand-induced EGFR phosphorylation [68].

There is evidence that GM3 regulates insulin signaling by lateral association with InsR, and thereby triggers development of insulin resistance. GM3 is the major ganglioside component of adipose tissue and was therefore expected to play a role in insulin signaling for regulation of metabolism in adipocytes. Our early studies demonstrated that TNF-α-induced insulin resistance in mouse adipocytes was concomitant with elevated GM3 content resulting from upregulated expression of GM3S gene, *St3gal5* [69]. Treatment of these cells with D-threo-1-phenyl-2-decanoylamino-3-morpholino-1 propanol (D-PDMP) [70], an inhibitor of glucosylceramide synthase (GCS) which depletes cellular GSLs (including GM3), normalized the defect in insulin-dependent Tyr phosphorylation of IRS-1, even in the presence of TNF-α. Consistently with these findings, GM3S KO mice displayed enhanced insulin signaling and were protected from HFD-induced insulin resistance [71]. We used a combination of analytical techniques (immunoprecipitation, cross-linking, and fluorescence recovery after photobleaching (FRAP)) in living cells, to demonstrate that the increased GM3 level in the state of insulin resistance dissociated InsR from caveolin-1 [72]. Caveolin-1, the resident coat protein of caveolae, localizes and stabilizes InsR in the microdomain, and under normal conditions, is required for proper insulin signaling in adipocytes. In the presence of TNF-α, increased GM3 competes with caveolin-1 for binding to InsR, resulting in dissociation of InsR/caveolin-1 complex. Mutation of a residue of the basic amino acid lysine (Lys^944^) located just above the transmembrane domain of InsR disrupted InsR/GM3 interaction. Analogously, mutation in EGFR of similarly located membrane proximal Lys^642^ resulted in loss of interaction with GM3 [61].

## 5. Ganglioside-Deficient Model Mice and Human Subjects

Gangliosides are expressed in all vertebrate tissues but are most abundant in nerve tissues. GM1, GD1a, GD1b, and GT1b are the predominant gangliosides in mammalian brains [2]. Mice that lack *Ugcg* gene-encoded GCS also lack the ganglioside synthetic pathway (Figure 1) and are embryonic-lethal [73]. Two groups independently studied mice with neuronal deletion of *Ugcg* using nestin-cre (*Ugcg* cKO). Gröne's group reported that neuronal cKO mice were born normally but lacked all GlcCer-based GSLs. Shortly after birth, these mice displayed severe ataxia with dysfunction of cerebellum and peripheral nerves and died within 24 days [74]. Proia's group reported that a different line of neural *Ugcg* cKO mice had a much milder phenotype, probably because they retained a low level of GSL expression. These mice displayed abnormal neurologic phenotype and severe loss of Purkinje cells within three months [75]. Another study of mice with Purkinje cell-specific *Ugcg* deletion showed that GlcCer-based GSLs are essential for axonal homeostasis and normal myelin sheath formation [76]. Oligodendrocyte-specific *Ugcg* deletion caused no evident abnormalities [77]. These findings, taken together, demonstrate that axonal GSLs (particularly gangliosides) in mammals are essential for neuronal function and axon/myelin interactions at various developmental stages.

Mice with deletion of GM2S (*B4galnt1*), whose ganglioside expression is limited to GM3 and GD3, appear normal when young, but display decreased myelination and progressive axonal degeneration in both central and peripheral nervous systems as they age [78,79]. Myelin-associated glycoprotein (MAG) is the binding partner of axonal gangliosides. MAG, also known as Siglec-4, is a member of the Siglec (sialic-acid binding immunoglobulin-type lectin) family and is expressed on the innermost myelin layer in oligodendrocytes and Schwann cells. GD1a and GT1b are selectively recognized by MAG through their Neu5Acα2-3Galβ1-3GalNAc sequence [80,81]. MAG KO mice with mixed genetic background, in 1995 and 2001 studies, developed normal myelin sheaths, but older individuals showed disruption of axon-myelin integrity and development of neuropathy [82,83]. In a 2005 study using the same MAG KO mice backcrossed to >99% C57BL/6 strain purity, phenotypes were more severe, with major axonal degeneration in central and peripheral nervous systems [84]. Phenotype of GM2S KO mice, which lack MAG-binding trisaccharide, was similar to that of MAG KO mice. GM2S/MAG-double KO mice had neuropathological and behavioral deficits similar to those of MAG KO and GM2S KO mice, indicating that MAG-ganglioside binding is essential for myelin/axonal integrity.

In view of the phenotype of neuronal *Ugcg* mutant mice that lack gangliosides in neurons, GM3S (*St3gal5*) KO mice might be expected to have similar defects. GM3S KO mice displayed enhanced insulin signaling, but did not have overt neurologic abnormalities, perhaps because of an alternative synthetic pathway for o-series gangliosides [71]. Studies by Schnaar's group showed that o-series gangliosides GM1b and GD1α expressed in GM3S KO brain contained MAG-binding sequence Neu5Acα2-3Galβ1-3GalNAc at their termini and bound to MAG [80,81], supporting the concept that alternatively expressed gangliosides compensate for loss of GD1a and GT1b in this case. In a 2009 study, our group demonstrated hearing loss resulting from degeneration of the organ of Corti in GM3S KO mice [85]. GM3 is the major ganglioside component of cochlea (but not in the central nervous system) and is essential for structural integrity and function of cochlear hair cells [86]. Within humans, mutations of the same *ST3GAL5* gene are associated with deafness.

GM3S/GM2S double KO (DKO) mice lack all ganglio-series gangliosides. They have small brains, develop severe neurodegenerative disease, and usually die by age 2.5 months [87]. Histopathological analysis of brains of these mice revealed notable vacuolar pathology in white matter regions, with axonal regeneration and disrupted interaction with myelin. Phenotype of DKO mice is less severe than that of neuronal *Ugcg* KO mice, perhaps because LacCer (a common precursor of other GSLs, including globosides) is still expressed. These findings support the concept that gangliosides play essential roles in axon/myelin interaction and other neuronal functions.

GD3S (*St8sia1*) KO mice, which lack b-series gangliosides, do not display overt clinical pathology, but have more subtle symptoms, such as reduced regeneration of axotomized hypoglossal nerves, thermal hyperalgesia, mechanical allodynia, and reduced response to prolonged noxious stimulation [88,89,90]. To establish a mouse model with further restriction of ganglioside expression, Proia’s group crossed GD3S KO mice with GM2S KO mice. The resulting DKO mice expressed only GM3. They appeared normal at birth, but soon underwent early-onset neurodegeneration and sudden lethal audiogenic seizures [90]. In studies of a similar DKO model, Furukawa's group found that mice did not undergo seizures but had reduced sensitivity of sensory nerves resulting from nerve degeneration, with consequent over-scratching and skin lesions [91]. These DKO mice also displayed dysregulated complement activation, with consequent inflammation and neurodegeneration [92].

LacCerS is encoded by two β4-galactosyltransferase genes, *B4galt5* and *B4galt6*, and complete LacCerS KO mice were established fairly recently. *B4galt5* KO mice show early embryonic lethality because of extra-embryonic defects, whereas *B4galt6* KO mice appear normal at birth and subsequent growth stages [93,94,95]. Neuron-specific *B4galt5* KO mice generated using nestin-cre (*B4galt5* cKO) also had apparently normal growth. In 2018, *B4galt5/B4galt6* DKO mice were generated by crossing *B4galt5* cKO and *B4galt6* KO mice [96]. These DKO mice had a phenotype similar to that of neuronal *Ugcg* cKO mice: they were born alive but displayed retarded growth, motor deficits at 2 weeks, and death by 4 weeks. LacCerS activity was absent in DKO brain, and in *B4galt5* cKO and *B4galt6* KO brains, had levels roughly half that of control brain, indicating that LacCerS activity in a mouse brain is dependent on both *B4galt5* and *B4galt6* genes. Axonal and myelin formation were strongly impaired in the DKO mice, presumably because MAG-binding gangliosides were completely absent from axons. Neuronal cell maturation and perineuronal net formation were also impaired in the cerebral cortex of DKO mice, presumably because of an absence of ganglioside/laminin interaction. Laminin/GM1 interaction plays a key role in nerve growth factor signaling: laminin binds directly to GM1 and induces clustering of GM1, TrkA, and β1 integrin [97]. Both GM2S KO and GM3S KO mice lack GM1, but do not display a severe neurologic phenotype like that of neuronal *Ugcg* cKO or *B4galt5/B4galt6* DKO mice. A possible explanation is that other gangliosides expressed in GM2S KO or GM3S KO mice can bind to laminin, compensating for the absence of GM1.

In humans, congenital disorders of ganglioside synthesis are extremely rare. To date, only mutations in *ST3GAL5* (GM3S gene) and *B4GALNT1* (GM2S gene) have been reported. *B4GALNT1* mutations occur in subjects with hereditary spastic paraplegia (HSP), a group of inherited neurodegenerative disorders characterized by progressive spasticity and weakness of the legs [98,99,100]. In complex forms of HSP, *B4GALNT1* mutation subjects also display mild to moderate intellectual disability, and in some cases, seizures, which are also observed in GM2S KO mice. Most cases of *B4GALNT1* mutation in humans involve complete loss of GM2S activity. GM2S KO mice may therefore provide a useful model of HSP [101].

*ST3GAL5* mutation subjects generally display severe defects relative to the mild neurological disorders observed in *B4GALNT1* mutation subjects. Most of the *ST3GAL5* mutation subjects are blind, deaf, intellectually impaired, and suffer from infantile-onset severe seizures [86,102,103,104,105,106,107]. In contrast, GM3S (*St3gal5*) KO mice are deaf but do not display seizures or neurological deficits. Total amount of gangliosides expressed in GM3S KO brain is maintained through increase of o-series species (GM1b, GD1α), and this o-series increase may compensate for loss of other major gangliosides (see paragraph 3 of this section). Whether brains of human subjects with a *ST3GAL5* mutation express levels of o-series gangliosides comparable to those in GM3S KO mice is yet to be determined.

## 6. Leptin Signaling in Ganglioside-Deficient Mouse Models and Cell Lines

Functional roles of gangliosides in LepR signaling were first investigated using mice with inducible deletion of GCS. To examine the role of neuronal GSLs in regulation of energy homeostasis, tamoxifen-inducible neuron-specific *Ugcg* KO mice were generated by crossing floxed *Ugcg* mice and CamK-CreERT2 mice, to avoid the lethality of systemic KO or developmental defects in previously reported neuronal KO [108]. Coincident with neuronal depletion of GSLs at 3 weeks post-induction (weeks p.i.), inducible, conditional KO (icKO) mice displayed a progressive increase of body weight and fat mass. They showed minor hyperphagia at 3 weeks p.i. (which disappeared by 6 weeks p.i.), as well as hypometabolism and hypothermia. LepR signaling in hypothalamic ARC neurons of icKO mice was reduced, as demonstrated using peripherally administered leptin-induced Stat3 phosphorylation or c-fos protein expression as indicators of neuronal activity. Interaction between GM1/GD1a and LepR in hypothalamic neuronal cell line N-41 was documented by an in situ proximity ligation assay and co-immunoprecipitation experiments. Depletion of GlcCer-derived gangliosides in these cells by treatment with n-butyldeoxynojirimycin (NB-DNJ), a specific inhibitor of GCS, strongly reduced leptin-induced JAK phosphorylation. These findings, taken together, indicate that leptin activity in hypothalamic neurons is enhanced by interaction of a-series gangliosides with LepR in the neuronal membrane.

Serum sample analysis of GD3S KO mice by Furukawa's group in 2015 further demonstrated the importance of ganglioside/leptin interactions [109]. Leptin levels at age 15 and 60 weeks were much lower in KO mice than in wild-type (WT) mice, although the KO mice had normal (not obese) body weight. Immunohistochemical analysis revealed accumulation of leptin in adipose tissues, indicating impaired leptin secretion. Consistent with this finding, analysis of primary culture of adipose-derived stromal vascular fractions (SVF) from KO mice revealed high leptin levels (relative to WT mice) in cell lysates and low levels in culture medium. This was not the case for adiponectin, another important adipocyte-derived hormone. Expression of a-series (GM3, GM1, GD1a) and b-series gangliosides (GD1b, GT1b) by SVF cells from WT was demonstrated by flow cytometry. Leptin secretion was restored by addition of exogenous GD3, GD1b, or GT1b (but not GD1a) to culture medium of SVF cells from GD3S KO mice. Leptin secretion was impaired by treatment of 3T3-L1 adipocytes with methyl-β-cyclodextrin, which depletes membrane cholesterol and disrupts lipid rafts. A strong shift of raft markers caveolin-1 and flotillin-1 from lipid rafts to non-rafts in white adipose tissue (WAT) of GM3S KO mice was revealed by sucrose density-gradient ultracentrifugation. Colocalization of leptin and caveolin-1 was observed by immunostaining in WT adipocytes, but not in GD3S KO adipocytes. Exogenous addition of GT1b to GD3S KO-derived cells restored such colocalization. The above findings clearly indicate that b-series gangliosides regulate leptin secretion from adipocytes in lipid rafts, however, the mechanism is unknown.

In a follow-up 2016 study, Furukawa's group described altered leptin signaling in the hypothalamus of GD3S KO mice, which may explain why these mice do not develop obesity despite greatly reduced levels of circulating leptin [110]. GM1 and GD1a levels in hypothalamus of GD3S KO were much higher than in WT, whereas major b-series gangliosides GD1b and GT1b disappeared. Hypothalamic mRNA and protein levels of LepRb were higher in GD3S KO than in WT. In association with upregulation of LepRb, basal levels of hypothalamic phospho-STAT3 (p-STAT3) were also increased in GD3S KO, and were further enhanced by leptin administration; consequently, STAT3 was more strongly activated in GD3S KO than in WT. GD3S overexpression in N-41 cells (which express mainly a-series gangliosides) results in a reduction of leptin-induced STAT3 phosphorylation, enhanced expression of b-series gangliosides, and associated decline of a-series gangliosides. Leptin stimulation-dependent interaction of LepR with GM1 or GD1a was demonstrated by co-immunoprecipitation. These findings indicate that increased levels of a-series gangliosides GM1 and GD1a in GD3S KO hypothalamus enhance leptin signaling, and thus compensate for reduced leptin secretion.

Studies by our group and Proia’s showed that body weight of GM3S KO mice subjected to HFD did not significantly differ from that of WT (C57BL/6 genetic background), although the KO mice displayed reduced insulin resistance and chronic low-grade inflammation [71,111]. In contrast, we found striking differences in obese phenotype of GM3S KO vs. WT of KK-*Ay* genetic background (characterized by severe and earlier onset of obesity and diabetic pathology) [112]. Whereas WT KK-*Ay* were hyperphagic and developed severe obesity, KK-*Ay*/GM3S KO had significantly lower body weight and food intake, and greater glucose and insulin tolerance. Hypothalamic response to peripheral administration of leptin, assessed by c-fos immunoreactivity, declined greatly by age 10 weeks in WT KK-*Ay* because of development of leptin resistance, but was still strongly present in KK-*Ay*/GM3S KO at this age. GM3S KO in N-41 cells, which lack a-series gangliosides and instead express the usually minor o-series ganglioside GM1b, resulted in reduced leptin-dependent STAT3 phosphorylation. This finding is consistent with that from the NB-DNJ-treated N-41 cells described in the first paragraph of this section. In contrast, GM3S KO cells showed strong enhancement of leptin-dependent ERK phosphorylation. Leptin-induced c-fos expression is controlled by activation of the SHP2-ERK pathway [16]; thus, the above findings suggest that the LepRb-ERK pathway is enhanced in GM3S KO mice and provides protection from leptin resistance.

## 7. Conclusions

The importance of gangliosides in leptin signaling is clearly documented by studies using various ganglioside-deficient mouse strains and cell lines (Table 1).

A-series gangliosides GM1 and GD1a positively regulate the LepRb-STAT3 pathway, as first indicated in studies of GCS icKO mice [108]. A subsequent study using the same GCS icKO mice and GCS inhibitor-treated hypothalamic cells suggested that GD1a negatively regulates InsR signaling and InsR protein levels in the hypothalamus, in contrast to LepRb signaling [113]. GCS icKO mice also displayed impairment of fasting-induced lipolysis and associated reduction of norepinephrine content in WAT. The regulatory mechanism of this process remains to be elucidated; however, these findings again reflect the variability of mode of activity by gangliosides on individual receptors. Leptin and insulin act together on hypothalamic neurons, and co-activation of the two pathways within ARC neurons is essential for the promotion of WAT browning and energy expenditure [114]. Cowley's group observed, in a recent study of DIO mice, that leptin signaling in ARC neurons (most likely AgRP neurons) blocked the suppressing effect of insulin on hepatic glucose production through upregulation of PTP1B, leading to hyperglycemia [115]. Proper expression of hypothalamic gangliosides appears to be essential for maintaining insulin- and leptin-mediated metabolic responses.

In contrast to LepRb-STAT3 signaling, LepRb-ERK signaling was enhanced in GM3S KO cells, in which o-series GM1b is overexpressed, and LacCer is accumulated in association with loss of a-series gangliosides. Which GSL(s) is involved in the ERK pathway, and whether this GSL promotes or inhibits the signal, remains to be determined. WT N-41 cells express a-series gangliosides GM3, GM2, GM1, and GD1a, whereas N-41/GM3S KO express GM1b as sole ganglioside species and have 70–80% lower total ganglioside content than the WT. Loss of a-series ganglioside(s), and/or increase of GM1b, may enhance LepRb-ERK signaling in GM3S KO. One possibility is that one (or several) of these gangliosides interacts directly with LepRb and subsequently enhances the ERK signal by modulating dimerization or conformation of the receptor, while such change acts in an opposite manner to concomitantly inhibit STAT3 signaling. An alternative possibility involves reported crosstalk mechanisms between LepR and other receptors (e.g., EGFR, IGF1R) [116,117,118,119]. EGFR and IGF1R are negatively regulated by GM3-related gangliosides, therefore, loss of these gangliosides may enhance ERK phosphorylation through transactivation of these receptors by leptin, in combination with activation by LepRb signaling.

In conclusion, gangliosides play essential roles in modulation of signals by LepRb and possibly several other receptors involved in energy homeostasis, in a ganglioside species-specific manner. The molecular mechanisms underlying these effects remain to be elucidated, however, direct interaction between LepRb and particular gangliosides, and/or receptor crosstalk between LepRb and other receptors in hypothalamic neurons, are strong possibilities. Further studies will clarify the mechanisms responsible for ganglioside-dependent control of energy balance.

## Figures and Tables

**Figure 1 ijms-21-05349-f001:**
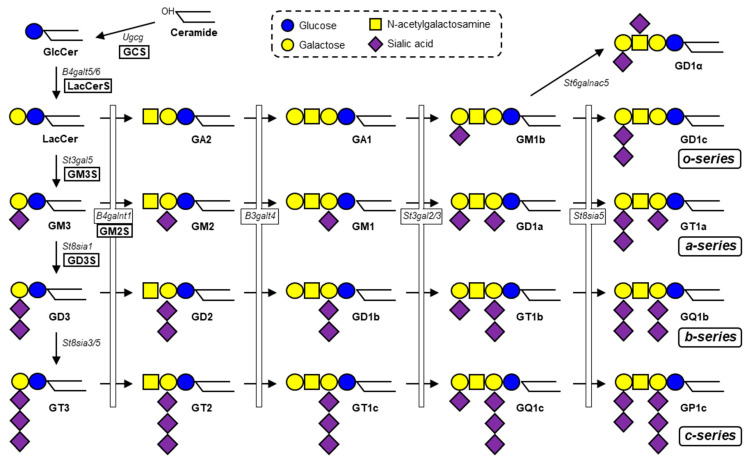
Biosynthetic pathway of ganglio-series gangliosides. GCS (*Ugcg*), a glucosyltransferase, catalyzes the first step in synthesis of ganglio-series gangliosides. Subsequently, LacCerS (*B4galt5/6*) adds a galactose residue onto GlcCer to form LacCer. GM3S (*St3gal5*) is a sialyltransferase required for initiation of synthesis of a- and b-series gangliosides. GD3S (*St8sia1*) is a sialyltransferase required for synthesis of b-series gangliosides. Four species (GM1, GD1a, GD1b, GT1b) comprise the majority of total brain gangliosides in mammals.

**Figure 2 ijms-21-05349-f002:**
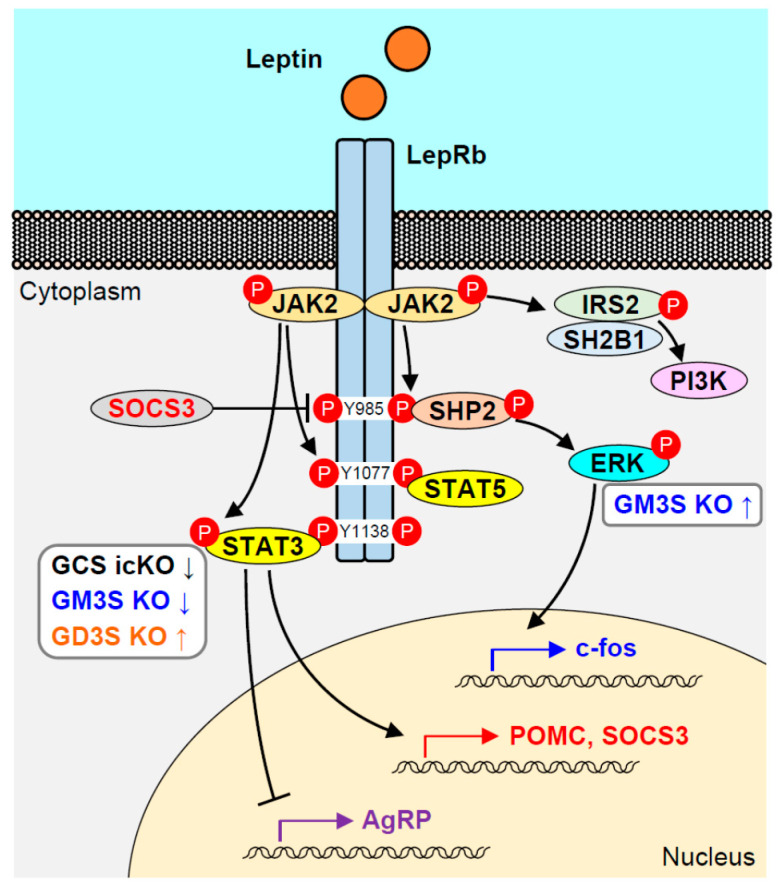
Leptin receptor signaling pathway, and alterations in signaling that characterize various ganglioside-deficient KO mouse models. Up and down arrows indicate increased or decreased activation of STAT3 or ERK in the pathways for the models. GCS icKO: tamoxifen-inducible, neuron-specific, conditional GCS KO.

**Table 1 ijms-21-05349-t001:** Major gangliosides expressed in mouse brains and N-41 cells with genetically or pharmacologically modified GSLs and their leptin signaling.

Model	Major Gangliosides	Leptin Signaling, in Comparison with That in WT	Reference
WT mouse brain	GM1, GD1a, GD1b, GT1b	-	
N-41 cells	GM3, GM2, GM1, GD1a	-	
GCS icKO brain	40–60% depletion	Reduced p-STAT3	[108]
NB-DNJ-treated N-41	80–90% depletion	Reduced p-STAT3	
GD3S KO brain	GM1, GD1a	Enhanced p-STAT3	[110]
GD3S-OE N-41	GD3, GD1b	Reduced p-STAT3	
GM3S KO brain	GM1b, GD1α	Enhanced c-fos expression	[112]
GM3S KO N-41	GM1b *	Reduced p-STAT3, Enhanced p-ERK	

NB-DNJ, n-butyldeoxynojirimycin; OE, overexpressed. * Total amount of gangliosides was lower than that in WT N-41 cells.

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
