# Peer review of "Roles of Gangliosides in Hypothalamic Control of Energy Balance: New Insights"

_ijms, 2020, doi:10.3390/ijms21155349_

Round 1

Reviewer 1 Report

In this manuscript, Inamori and Inokuchi revise the essential role of gangliosides in hypothalamic control of energy balance through the modification of LepR downstreaming pathway.

The topic is of high interest due to the possible use of ganglioside as regulator of energy balance although the molecular mechanisms have yet to be finely described.

I recommend publication of this manuscript 

Minor point

Introduction section, lines: 54-73. I suggest to move this paragraph to the following section “Leptin receptor signaling”

Author Response

Reviewer 1: Introduction section, lines: 54-73. I suggest to move this paragraph to the following section “Leptin receptor signaling”

Response: Thank you for the suggestion. We have amended accordingly.

Reviewer 2 Report

This is an excellent review of the literature on the role of gangliosides in the regulation of receptor tyrosine kinases, summarizing the results obtained using KO mouse models revealing the essential role of gangliosides in hypothalamic control of energy balance by modulating the effects of leptin.

The review is well written and perfectly up to date.

I have only few comments that the authors could take into account to improve the manuscript:

  1. Figure 1: Biosynthetic pathway of ganglio-series gangliosides: the scheme is incomplete; the c-series gangliosides are missing. This can be a problem for non-specialist of GSLs, even if the authors do not referred to c-series in the text. At least, the authors should indicate that the scheme is simplified. In parallel, the authors should apply the recommendations of the Consortium for Functional Genomics for gangliosides’ drawing.
  2. The chapter summarizing the different signaling pathways that regulate the effects of leptin is a bit too long and could be reduced.
  3. The use of semicolons is sometimes inappropriate and should be avoided.

Author Response

Reviewer 2:

Figure 1: Biosynthetic pathway of ganglio-series gangliosides: the scheme is incomplete; the c-series gangliosides are missing. This can be a problem for non-specialist of GSLs, even if the authors do not referred to c-series in the text.

Response: Thank you for pointing out that the c-series was missing in the pathway. We have revised Figure 1 to include the c-series gangliosides.

The use of semicolons is sometimes inappropriate and should be avoided.

Response 2: Thank you for pointing out inappropriate use of semicolons. We have corrected the errors accordingly (please see the Track Changes in the revised manuscript).